# From Point Annotations to Epithelial Cell Detection in Breast Cancer Histopathology using RetinaNet

**Caner Mercan**                                        Caner.Mercan@radboudumc.nl
**Maschenka Balkenhol**                          Maschenka.Balkenhol@radboudumc.nl
**Jeroen van der Laak**                        Jeroen.vanderLaak@radboudumc.nl
**Francesco Ciompi**                                Francesco.Ciompi@radboudumc.nl
*Computational Pathology Group, Radboud University Medical Center, Nijmegen, the Netherlands*

**Editors:** Under Review for MIDL 2019

## Abstract

Detection of epithelial cells has powerful implications such as being an integral part of nuclear pleomorphism scoring for breast cancer grading. We exploit the point annotations inside nuclei boundaries to estimate their bounding boxes using empirical analysis on the cell bodies and the coarse instance segmentation masks obtained from an image segmentation algorithm. Our experiments show that training a state-of-the-art object detection network with a recently proposed optimizer on simple bounding box estimations performs promising epithelial cell detection, achieving a mean average precision (mAP) score of 71.36% on tumor and 59.65% on benign cells in the test set.

**Keywords:** Epithelial cell detection, RetinaNet, breast histopathology.

## 1. Introduction

In whole slide breast histopathology image analysis, tumor detection is considered to be the fundamental step for scoring nuclear pleomorphism which is one of the three main components of breast cancer grading, along with mitotic count and tubule formation. The morphology of epithelial cells changes as the cancer arises, nuclei become more atypical in structure and internal organization. One of the challenges of tumor detection arises from this structural irregularity of tumor cell morphology. The nuclei can have arbitrary shapes and variable sizes, making tumor detection a difficult task. Another challenge arises from the lack of pixel-wise fine-grained tumor annotations. The annotation process becomes too time-consuming for pathologists due to the large number of erratic cells.

In this paper, we tackle the detection and the binary classification of tumor vs. benign cells in whole slide breast histopathology images from only point annotations in the nuclei of epithelial cells. We propose an epithelial cell detection and classification method involving a state-of-the-art deep network for object detection that is suitable for dense images with large number of objects. The object detection network requires images to have outlined objects in the form of rectangular bounding boxes. Additionally, each outlined object in the image is associated with a single label. Due to the steep cost of high-quality pixel-wise annotations of bounding boxes, we utilize point annotations placed on the nuclei to estimate the corresponding bounding boxes. In this regard, we investigate two different methods to estimate each nucleus structure from which we obtain its bounding box.

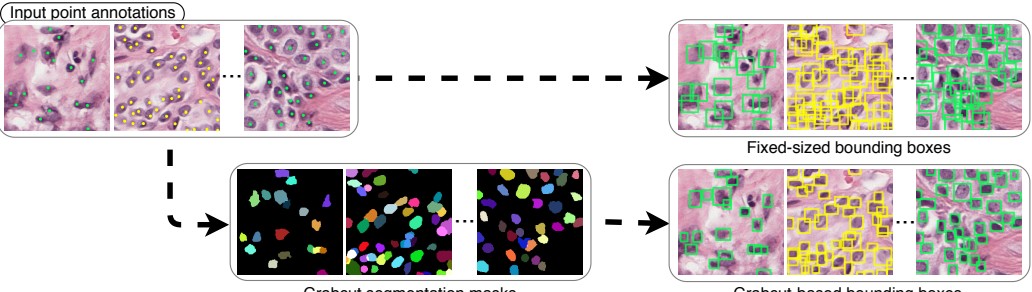

Figure 1: Comparison of bounding box estimations. Green denotes tumor and yellow denotes benign cells.

## 2. From point annotations to epithelial cell detection and classification

RetinaNet (Lin et al., 2017) is one of the state-of-the-art single-shot object detection and classification algorithms that is robust against imbalanced class distribution and it has superior performance on dense images with large number of objects. Therefore, this architecture is our choice of object detection network for the task of epithelial cell detection and classification in breast histopathology images. However, training RetinaNet for the task of binary classification of epithelial cells as either benign or tumor requires a bounding box annotation of the nuclei. For this purpose, we investigate two approaches; the first is using a fixed-sized bounding box from empirical observation of the cell structure and the second is estimating a segmentation-based rectangular bounding box for each nucleus. The former involves a bounding box with fixed-sized width and height around the point annotations whereas the latter investigates Grabcut (Rother et al., 2004) algorithm to coarsely segment the nuclei and then estimate the bounding box from the resulting instance segmentation mask.

For the first approach, we set the height and the width of the fixed-sized bounding boxes as a result of empirical analysis of the epithelial cells in the training set. The second approach involves Grabcut-estimated bounding boxes. We place two priors on the Grabcut algorithm for the estimation of an instance segmentation mask for each nucleus. Our first prior is a small mask around the point annotation denoting foreground presence and our second prior is a larger mask denoting a possible foreground presence. The mask sizes are determined based on the average epithelial cell size. The algorithm iteratively produces a new segmentation mask if the output mask is smaller than a threshold. We present the resulting bounding box estimations from both approaches in Figure 1.

## 3. Experimental setup and results

Our data set consisted of 39 whole slide breast histopathology images, each belonging to a different patient. We split the data into three sets of training, validation and test sets considering a uniform distribution of slides based on their nuclear pleomorphism scores. We extracted patches of $256 \times 256$ pixels from the slides at $40\times$ magnification. The patches involved bounding boxes annotated as either tumor or benign but they also included various types of cell bodies including but not limited to apoptotic and mitotic figures which were all regarded as background. Our experiments showed that using spatial data augmentations such as rotations, horizontal and vertical flips performed better compared to their combinations with color and noise data augmentation techniques.

Table 1: Classification results of the epithelial cells in the validation and the test sets for different combinations of bounding box estimators and RetinaNet optimizers (mAP in %).

| BoundingBox | Optimizer | Validation set | | Test set | |
|---|---|---|---|---|---|
| | | mAP (tumor) | mAP (benign) | mAP (tumor) | mAP (benign) |
| Grabcut | Adam | 72.75 | 40.28 | 57.27 | 45.04 |
| Fixed | Adam | 75.76 | 62.72 | 67.55 | 61.55 |
| Fixed | AdaBound | 79.90 | 59.95 | 71.36 | 59.65 |

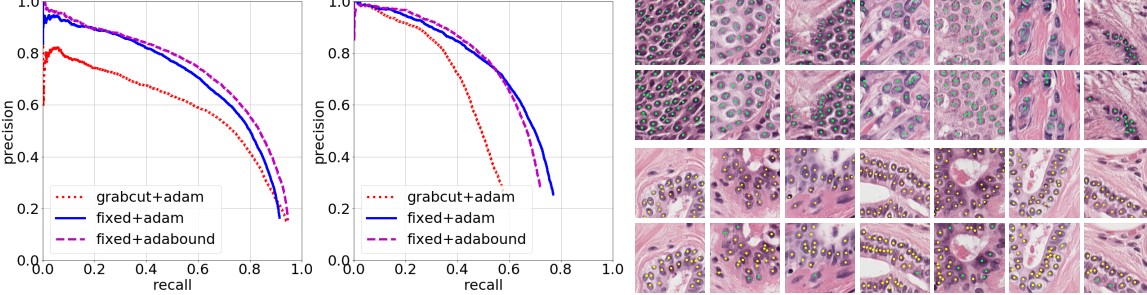

Figure 2: Precision-recall curves of the methods on tumor (left) and benign cells (right) in the test set.

Figure 3: Cell detection results with ground truth (top) and predictions (bottom).

We investigated RetinaNet using different ResNet backbones and found out that ResNet50 provided sufficient depth for the best classification performance. We used the ImageNet pre-trained weights to initialize the network and tweaked the last layer to perform two-class classification. Apart from the usual Adam optimizer, we utilized AdaBound (Luo et al., 2019); a novel optimizer combining the fast convergence properties of Adam during the early stages of training and promising better loss like SGD after several epochs.

We present the quantitative classification results in Table 1 and in Figure 2. The bounding boxes obtained through the segmentation masks of the Grabcut algorithm suffered due to the imperfections in the masks whereas the fixed-sized bounding boxes generally covered the nuclei body better. In addition, the Adabound optimizer outperformed Adam, showing better tumor detection and classification performance, 71.36% vs. 67.55%, with the cost of slightly worse performance in benign cells, 59.65% vs. 61.55%. Note that the number of tumor annotations in the data set was roughly three times the number of benign cell annotations, therefore better performance in tumor contributed to the overall performance more. Finally, we present some visual results from the best model on the patches in the test set in Figure 3.

## 4. Conclusion

We presented a powerful application of one of the state-of-the-art object detectors for the detection and classification of epithelial cells in breast cancer using simple point annotations. The combination of fixed-sized bounding box estimations and RetinaNet with AdaBound optimizer trained on spatially augmented patches performed the best in our experiments. For future work, we will investigate different training schemes to improve the false positive rate for tumor and the detection model will be used as a building block to score nuclear pleomorphism for breast cancer grading.

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

## Appendix A. Slide-level detection results

We present the slide-level predictions on three slides in the test set in Figure 4.

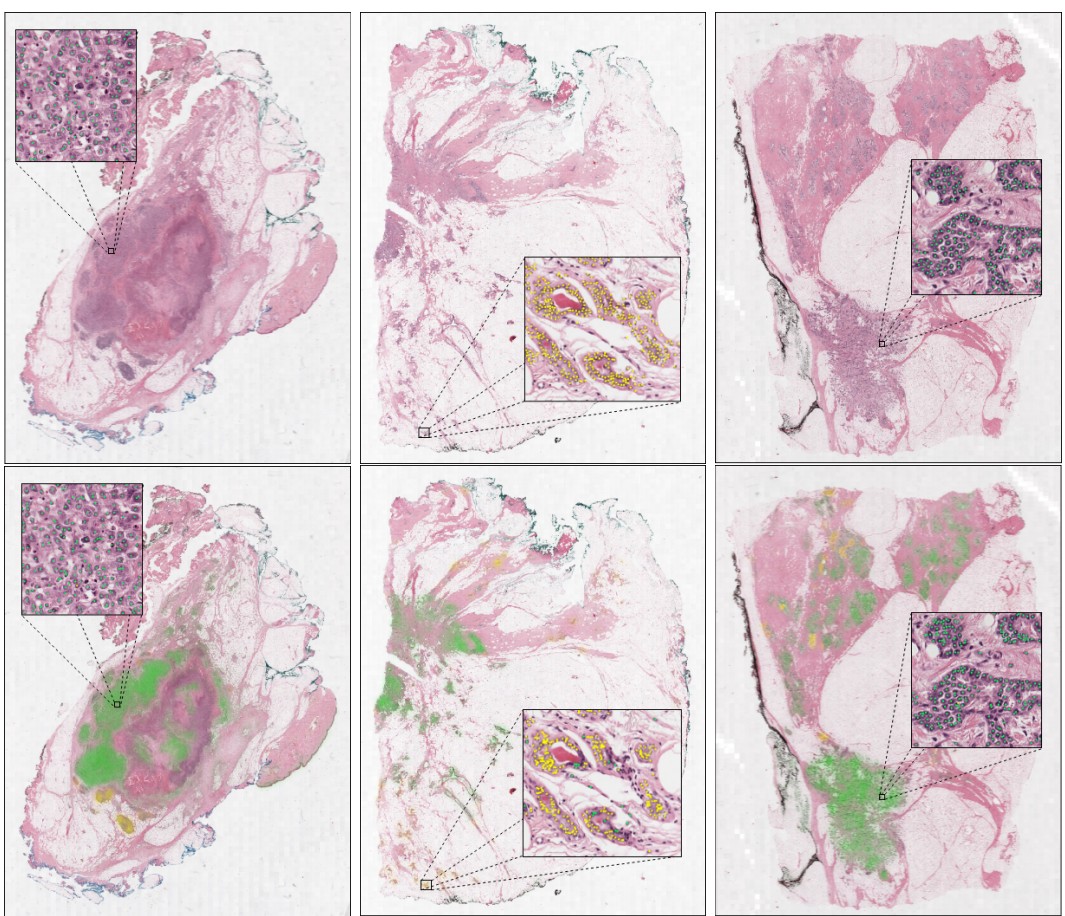

Figure 4: Input whole slide images (top) and the corresponding predictions (bottom). Example small regions are highlighted to better compare the ground truth annotations and the predictions visually.

