# OpenReview forum: "From Point Annotations to Epithelial Cell Detection in Breast Cancer Histopathology using RetinaNet"
_MIDL.io/2019/Conference/Abstract — MIDL Abstract 2019_

### Official Review · AnonReviewer2 · 2019-04-29
**Detection of epithelial cells in Breast cancer**

**Rating:** 3
**Confidence:** 2

**Review:**

The paper provides an object detection strategy for epithelial cell detection from breast cancer histopathology images. The method uses RetinaNet to classify regions obtained by point annotations as benign or tumor.

*Two types of bounding box estimation are investigated.
*Experiment with different optimizers like Adam or Adabound is illustrated.
*The paper is easy to understand.
*How are the bounding box obtained from grabcut standardized?
*Experimental results about data augmentation would also be interesting.

---

### Official Review · AnonReviewer1 · 2019-05-01
**Application-oriented abstract using RetinaNet to detect epithelial cells in breast cancer histopathology**

**Rating:** 2
**Confidence:** 3

**Review:**

The RetinaNet with different optimizers (Adam vs. AdaBound) and bounding boxes (Grabcut vs. Fixed) was trained, validated, and tested in this work on a total of 39 whole-slide breast histopathology images of 39 patients. The best methods achieved mean average precision scores of ~71.4% and ~59.7% on tumor and benign cells respectively.

Pros:

1. The paper addresses an interesting and challenging application.
2. In the methods the paper takes advantage of state-of-the-art methods.

Cons:

1. In this application-oriented paper, the state-of-the-art in epithelial cell detection and classification in breast cancer histopathology has not been discussed or reviewed. Therefore, the significance and impact of the work has not been well described.

2. It is not clear if the obtained precision scores are competitive in the field and if the proposed method can make a difference in state of the analysis of breast cancer. The scores are relatively low and the precision-recall curves, especially for Grabcut+Adam, for example, do not indicate a strong classification performance.

3. It was mentioned that the data was split to three sets of training, validation, and test, but not mentioned how. Was it split to equal sizes?

4. Figures 3 and 4 for visual assessment needed to be zoomed in for better evaluation.

5. Some important details of the networks and techniques are missing. The work should be reproducible.

---

### Decision · Program_Chairs · 2019-05-06
**Acceptance Decision**

Accept